# A Cross-Dataset Study for Text-based 3D Human Motion Retrieval

## Abstract

*We provide results of our study on text-based 3D human motion retrieval and particularly focus on cross-dataset generalization. Due to practical reasons such as dataset-specific human body representations, existing works typically benchmark by training and testing on partitions from the same dataset. Here, we employ a unified SMPL body format for all datasets, which allows us to perform training on one dataset, testing on the other, as well as training on a combination of datasets. Our results suggest that there exist dataset biases in standard text-motion benchmarks such as HumanML3D, KIT Motion-Language, and BABEL. We show that text augmentations help close the domain gap to some extent, but the gap remains. We further provide the first zero-shot action recognition results on BABEL, without using categorical action labels during training, opening up a new avenue for future research.*

## 1. Introduction

Dataset bias is a known phenomenon in machine learning research. The pioneering work of Torralba and Efros [26] shows that given a sample from an object recognition dataset, both a human researcher and a computer (SVM classifier) can guess which dataset the image comes from, known as the 'Name That Dataset' task. In a similar spirit, we observe that 3D human motion description datasets typically have a language style that distinguishes them from each other. KIT Motion-Language (KITML) [17] is dominated by locomotive motions and often starts by 'A person is...'. HumanML3D [8] similarly contains such full-sentence descriptions, but tends to be more verbose, and covers a larger vocabulary of motions. BABEL [18] language style is distinct, concisely describing with a single verb such as 'sit'. The t-SNE [27] visualization in Figure 1 confirms this observation, where we plot MPNet [22] text embeddings of random subset of 400 labels from each dataset. BABEL textual labels appear clearly distinct from HumanML3D and KITML. In this work, we perform cross-dataset evaluations to quantify these gaps, and attempt reducing them via text augmentations.

We instantiate our study with the text-to-motion retrieval task. While there is a large literature on text-to-motion

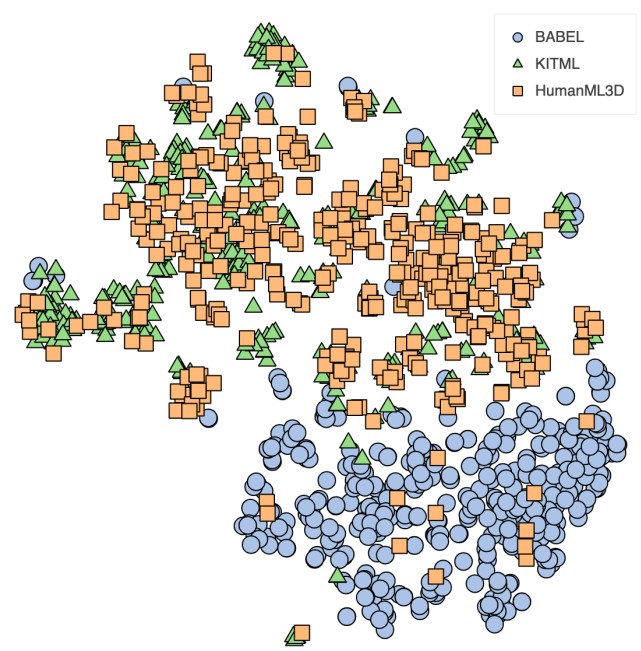

Figure 1. **3D human motion descriptions per dataset:** The t-SNE plot of text embeddings corresponding to motion descriptions clearly shows a domain gap between the concise raw labels of the BABEL dataset and the full-sentence labels of HumanML3D and KITML datasets.

synthesis [1, 2, 5, 7, 9, 10, 25, 30], text-to-motion retrieval is relatively new [8, 16, 29]. TMR [16] employs a contrastive training, similar to CLIP [20], to learn a cross-modal embedding space. In this work, we train TMR models and show several improvements. Similar to ActionGPT [11] which improves text-to-motion synthesis with text augmentations, we leverage large language models (LLMs) to increase robustness of retrieval models via label augmentations such as paraphrasing (see Table 1). Furthermore, we study the ability of a model trained with free-form text labels to generalize to the zero-shot[1] action recognition task, by performing motion-to-text retrieval.

Our contributions are the following: (i) We report

---

[1]Similar to contemporary literature [20], we abuse the term zero-shot, meaning training on a separate dataset than the downstream dataset used for evaluation.

| Original label | Paraphrases | Action |
|---|---|---|
| A person stumbles on the ground but gets up and keeps on running. | -Someone trips and falls but continues moving forward by getting up and running.
-An individual experiences a misstep while running but continues onward.
-A person stumbles while running but gets back up and continues to move forward. | Trip and run. |
| A person knees on the floor. | -A person is crouching or squatting on the ground.
-Someone is bending their knees to lower themselves to the ground.
-The individual is kneeling on the ground. | Kneel. |
| Punch. | - A person clenches their fist and strikes something with the closed hand, using the arm and shoulder muscles for force.
-A person extends their arm and fist in a punching motion.
-A person thrusts one fist forward then pulls it back. | N/A |

Table 1. **Example LLM paraphrasing:** We prompt Llama-2 as described in Section 3 in order to augment the original motion descriptions on the left. Middle column shows results of instructing the LLM to paraphrase. The right column is the result of instructing to convert into the style of action labels. The three example input labels are taken from HumanML3D, KITML, and BABEL datasets from top to bottom.

cross-dataset retrieval performance using TMR on a unified SMPL [12] representation, and assess the effect of training on a combination of datasets, leveraging HumanML3D, KITML and BABEL. (ii) We perform data augmentation on the textual labels and show that training TMR with these augmented data improves the results. (iii) We perform zero-shot action recognition on the BABEL-60, BABEL-120 benchmarks by training only on HumanML3D, and provide several ablations, again confirming the improvements from text augmentations.

## 2. Related Work

We briefly describe few works on (i) 3D human motions and language, with a particular emphasis on datasets in this domain, and (ii) zero-shot classification with natural language supervision in other domains of computer vision. For a broader overview, we refer to the survey of [31].

**3D human motions and language.** Following advances in natural language processing, there has been an increased interest in building models to control 3D human motion generation with language inputs [1, 2, 5, 7, 9, 10, 25, 30], and more recently on text-based motion search [8, 16, 29]. The performance of these models naturally depend on the datasets they are trained on. KITML [17] is one of the first 3D human motion description datasets, collecting annotations for a relatively small amount of motions, with a relatively small vocabulary of words, thus limiting its generalization to out-of-distribution samples. More recently, two concurrent works HumanML3D [8] and BABEL [18] collected manual labels for the large AMASS [13] motion collection. Since these efforts were in parallel, the resulting annotations differ in style, incurring a domain gap. As mentioned in Section 1, HumanML3D follows KITML-style verbose full sentence descriptions, while BABEL introduces concise labels, typically with verbs in an imperative form (e.g., 'wave hands' vs 'A person is waving hands'). In this work, we focus on a cross-dataset study investigating generalization performance of text-to-motion retrieval models, instantiated by the recent method of TMR [16].

In a similar spirit to our work, Action-GPT [11] investigates text augmentations using LLMs for improving robustness. However, their study is on a single dataset, BABEL, with only qualitative results on unseen text descriptions. Here, we provide quantitative cross-dataset results, showing improvements on the zero-shot setting with text augmentations.

**Zero-shot classification with natural language supervision.** CLIP [20] image-text retrieval model is a popular example of training contrastively with free-form language labels and successfully applying on categorical labels for zero-shot classification on various downstream datasets. CLIP observes a small performance gain by appending the string 'a photo of' to the class labels, simply to reduce the domain gap between training and test time. Similar multimodal contrastive models were built by ActionCLIP [28] for video action recognition, using additional prompts such as 'human action of'. In 3D human motions domain, MotionCLIP [24] leverages CLIP image-text joint space by turning 3D motions into rendered images. Similar to this work, MotionCLIP [24] reports results on BABEL action recognition benchmarks by posing the problem as motion-to-text retrieval; however, they work with the fully-supervised setting, where they use training labels of BABEL, adapting to the textual domain of action classes. In contrast, our target is the zero-shot setting, where the set of labels are unknown.

Figure 2. **Model overview:** We simply employ TMR [16] for text-motion retrieval, but unify several text augmentation approaches to increase its robustness across domains. For each ground truth (GT) textual label, we generate $n$ paraphrased versions, as well as a short action-style description using Llama-2 prompting. During training, we randomly sample either of these augmented labels with probabilities defined by $p_{gt}, p_{par}, p_{avg}, p_{act}$. With probability $p_{avg}$, we also randomly subsample paraphrased versions and average their text embeddings. The selected text embedding $z^T$ is then matched to the motion embedding $z^M$ using contrastive loss. Note that we do not visualize the motion decoder for simplicity, but we keep the original architecture as in [16].

## 3. Methodology

We build on the recent method of TMR [16], and make several improvements: mainly the use of text augmentations and using a hard-negative variant (HN-NCE [19]) of the InfoNCE [14] loss. We also train on a combination of datasets (instead of a single dataset) using motion representation of Guo et al. [8] computed on the SMPL [12] body skeletons (instead of dataset-specific skeletons). When training jointly on multiple datasets, we simply append training sets and sample disproportional to training set size to balance the distributions. In the following, we detail our text augmentation procedure.

We perform text augmentation by paraphrasing each motion text label several times. First, given a motion, for each of its text annotations, we use Llama-2 [6] to generate paraphrases of this text. We prompt Llama-2 by instructing to paraphase a given motion description with the paraphrasing style defined by few-shot examples that we provide in the form of text pairs. This procedure applies to HumanML3D and KITML sentences. When paraphrasing concise BABEL text annotations, we alter the prompt by instructing to describe the motion, and providing few-shot examples in the form of *"Sentence: 'Point.' Paraphrased: 'A person motions forward with their hand.' "*.

For HumanML3D and KITML, that are annotated with full sentences, we additionally generate action-style annotations. For example, an action-style annotation for *"The person sprints down the track, their feet pounding against the ground"* is *"Sprint"*. We refer to Table 1 for more text augmentation examples.

We have two sources for providing few-shot examples in the prompts. First, we generate example pairs using GPT-3.5 [15]. Second, we leverage the multiple annotations corresponding to the same motion segment (either within or across datasets), and assume that such annotations may be paraphrases of each other.

As a final augmentation strategy, we sample uniformly at random, among a set including all the annotations (ground truth

and its augmentations). We then encode all the texts in this set and average their associated text embeddings.

During training, for each motion in a batch, we sample with probability $p_{gt}$, one of the ground truth annotations (in case of multiple manual labels); with probability $p_{par}$, one of the paraphrased versions; with probability $p_{act}$, the action-style annotation version; and with probability $p_{avg}$, the averaged text embedding as described above. In our experiments, these are set as $p_{gt} = 0.4$, $p_{par} = 0.2$, $p_{act} = 0.1$ and $p_{avg} = 0.3$, unless stated otherwise. We illustrate this procedure in Figure 2.

## 4. Experiments

We first describe the datasets (Section 4.1) and evaluation metrics (Section 4.2) used in our experiments. Next, we report the main results on text-to-motion retrieval (Section 4.3) and zero-shot action recognition (Section 4.4). We then provide ablations on text augmentations (Section 4.5) and conclude with qualitative analyses (Section 4.6).

### 4.1. Datasets

We experiment with **HumanML3D** [8] and **KITML** [17] standard text-motion datasets. We also benchmark this task on **BABEL** [18] raw textual labels, and report on its action recognition benchmarks, BABEL-60 and BABEL-120 for 60 and 120 action labels, respectively. The source of these captioned motions largely overlap with the AMASS [13] collection that unifies motions from multiple MoCap sources into SMPL body format [12]. We therefore simply extract motion representation from Guo et al. [8] on SMPL skeletons for each of these datasets, alleviating the issue of dataset-specific skeleton definitions, e.g., for KITML [17].

HumanML3D includes 23384, 1460 and 4384 motions for the training, validation and testing sets, respectively. The original KITML dataset includes 6018 motions processed using the Master Motor Map (MMM) framework, split into sets of

| Training data | Augm | HN-NCE | HumanML3D | | | KITML | | | BABEL | | |
|---|---|---|---|---|---|---|---|---|---|---|---|
| | | | R@1 | R@3 | R@10 | R@1 | R@3 | R@10 | R@1 | R@3 | R@10 |
| H | ✗ | ✗ | $11.63_{\pm0.16}$ | $21.73_{\pm0.40}$ | $40.73_{\pm0.89}$ | $25.06_{\pm0.85}$ | $42.53_{\pm2.29}$ | $63.82_{\pm1.20}$ | $15.85_{\pm3.53}$ | $25.78_{\pm6.22}$ | $42.33_{\pm3.25}$ |
| K | ✗ | ✗ | $2.81_{\pm0.24}$ | $6.19_{\pm0.10}$ | $12.34_{\pm0.72}$ | $21.75_{\pm2.56}$ | $37.45_{\pm2.08}$ | $59.79_{\pm2.10}$ | $5.42_{\pm2.37}$ | $11.26_{\pm3.72}$ | $20.29_{\pm3.96}$ |
| B | ✗ | ✗ | $1.65_{\pm0.20}$ | $3.02_{\pm0.46}$ | $6.96_{\pm0.46}$ | $9.58_{\pm1.16}$ | $17.85_{\pm1.86}$ | $32.11_{\pm2.03}$ | $23.29_{\pm5.02}$ | $36.93_{\pm2.02}$ | $54.42_{\pm0.51}$ |
| H + K | ✗ | ✗ | $11.68_{\pm0.32}$ | $21.70_{\pm0.56}$ | $40.25_{\pm0.01}$ | $24.24_{\pm0.70}$ | $44.91_{\pm1.78}$ | $71.24_{\pm3.06}$ | $20.38_{\pm4.67}$ | $26.35_{\pm6.19}$ | $44.50_{\pm1.16}$ |
| H + K | ✓ | | $14.47_{\pm0.67}$ | $24.94_{\pm0.48}$ | $45.54_{\pm0.88}$ | $27.95_{\pm2.64}$ | $46.23_{\pm1.52}$ | $71.59_{\pm0.98}$ | $18.47_{\pm3.80}$ | $29.31_{\pm2.75}$ | $48.64_{\pm1.02}$ |
| H + K | | ✓ | $13.31_{\pm0.54}$ | $23.67_{\pm0.50}$ | $42.77_{\pm1.19}$ | $27.40_{\pm1.79}$ | $46.73_{\pm2.16}$ | $69.76_{\pm1.38}$ | $15.98_{\pm1.44}$ | $28.39_{\pm1.75}$ | $39.67_{\pm1.36}$ |
| H + K | ✓ | ✓ | $\mathbf{14.89}_{\pm0.77}$ | $\mathbf{26.34}_{\pm1.11}$ | $\mathbf{46.49}_{\pm0.50}$ | $29.39_{\pm1.82}$ | $46.82_{\pm2.44}$ | $68.96_{\pm1.09}$ | $14.68_{\pm2.32}$ | $29.86_{\pm5.50}$ | $42.07_{\pm4.39}$ |
| H + K + B | ✗ | ✗ | $10.02_{\pm0.43}$ | $19.37_{\pm0.13}$ | $37.48_{\pm1.02}$ | $22.46_{\pm2.22}$ | $42.68_{\pm1.21}$ | $66.35_{\pm1.21}$ | $26.34_{\pm2.31}$ | $41.42_{\pm5.26}$ | $57.08_{\pm0.93}$ |
| H + K + B | ✓ | | $12.25_{\pm0.11}$ | $23.31_{\pm0.02}$ | $42.38_{\pm0.23}$ | $24.30_{\pm1.65}$ | $46.89_{\pm1.46}$ | $71.62_{\pm0.64}$ | $24.80_{\pm6.94}$ | $39.03_{\pm5.32}$ | $56.90_{\pm0.70}$ |
| H + K + B | | ✓ | $11.53_{\pm0.47}$ | $20.48_{\pm0.48}$ | $38.39_{\pm0.64}$ | $26.04_{\pm0.26}$ | $46.39_{\pm1.93}$ | $71.33_{\pm0.32}$ | $26.37_{\pm3.34}$ | $41.47_{\pm4.17}$ | $55.69_{\pm1.09}$ |
| H + K + B | ✓ | ✓ | $12.38_{\pm0.57}$ | $23.66_{\pm0.36}$ | $44.05_{\pm0.72}$ | $26.63_{\pm3.25}$ | $47.16_{\pm1.86}$ | $72.06_{\pm0.84}$ | $28.47_{\pm1.80}$ | $39.80_{\pm0.69}$ | $56.45_{\pm2.46}$ |

Table 2. **Cross-dataset text to motion retrieval results:** We provide experiments on HumanML3D (H), KITML (K) and BABEL (B) datasets. Training on individual datasets perform worse than training on combined versions. Text augmentations (Augm) and HN-NCE loss overall improve results, especially on HumanML3D. We report the average across three training runs, together with the standard deviation denoted with $\pm$. Note we observe more stable results on HumanML3D compared to KITML and BABEL, on which we base our conclusions more safely.

4888, 300, 830 motions. The AMASS collection contains the majority of KITML motions, fitting SMPL body model to the corresponding MoCap markers, and therefore significantly differing in the skeleton definition. Due to imperfect intersection between AMASS and KITML (i.e., missing SMPL parameters for some KITML motions), our KITML dataset contains slightly less motions: 4688, 292 and 786 samples in the training, validation and testing sets, respectively. For BABEL, we use the official split, but we use the validation set for evaluation (as in other works on synthesis [3, 4]) given the absence of a publicly available test set. BABEL with text labels includes 64826 and 23734 motions for the training and testing sets; BABEL-60, 59834 and 22004; BABEL-120, 62650 and 22918. It is worth noting that, when training with a combination of datasets, we remove any sample that overlaps (in time) with a motion appearing in the evaluation set of any dataset.

As previously mentioned, the text annotations differ in length across datasets. We compute that the average number of words in original annotations are 12.4 for HumanML3D, 8.5 for KITML, and 2.3 for BABEL, confirming our observations. When paraphrasing, we generate 30, 30, 10 new annotations per sample for HumanML3D, KITML, BABEL labels, respectively.

## 4.2. Evaluation protocol

We report recall at several ranks as in [16] for both text-to-motion retrieval and action recognition (i.e., motion-to-text retrieval). Given an input modality, rank $k$ recall corresponds to the percentage of inputs whose label has been retrieved among the top $k$ results. For action recognition, we additionally report class-balanced accuracy (Top-1 CB), by averaging the Top-1 accuracies over action categories.

For the text-to-motion retrieval task, we report metrics using the 'All with threshold' protocol described in [16]. Within the test set, we compute the similarity across texts using their MPNet [23] embeddings. The rank of a sample is taken as the highest rank among the ranks of all its similar samples. We consider two samples to be similar if their text similarity is above 0.95. This protocol mitigates the performance artifacts that the large number of repeated or very similar text descriptions across motions could induce. As explained in [16], indeed, inside the retrieval gallery, a motion with a label very similar to the query text could wrongly be considered negative. With the 'All with threshold' protocol, it is considered a correct retrieved motion.

We run each training 3 times with different random seeds, and report the average results over these models. This is to account for the substantial fluctuations we observe when evaluating on KITML and BABEL text-to-motion benchmarks. For action recognition evaluation BABEL, we do not observe instability and report one training per experiment for simplicity.

## 4.3. Text-to-motion retrieval results

In Table 2, we report rank R@1, R@3 and R@10 metrics for text-to-motion retrieval, using protocol 'All with threshold' as described in [16]. We evaluate on HumanML3D, KITML and BABEL (raw text labels), comparing the performances of different training sets. As mentioned in Section 4.1, when cross validating, we remove from the training sets, motion segments that overlap with the testing sets of any of the datasets (even if the text labels are different).

**Cross-dataset evaluations.** In the first three rows of Table 2, we provide baseline trainings on individual datasets without any text augmentations. We see that KITML-only or BABEL-only training does not generalize to HumanML3D. On the other hand, HumanML3D-only training outperforms KITML-only training when evaluating on the KITML test set, which can be explained by the large size of HumanML3D, and both datasets having sentence-style labels. Unsurprisingly, BABEL label style being very different from the other two, BABEL-only training does not transfer well. We note that, upon observing instability on

| Method | Training data | Augm | BABEL-60 | | | BABEL-120 | | |
| --- | --- | --- | --- | --- | --- | --- | --- | --- |
| | | | Top-1 CB | Top-1 | Top-5 | Top-1 CB | Top-1 | Top-5 |
| 2s-AGCN [18, 21] CE | B-actions | ✗ | 24.46 | 41.14 | 73.18 | 17.56 | 38.41 | 70.49 |
| 2s-AGCN [18, 21] Focal | B-actions | ✗ | 30.42 | 33.41 | 67.83 | 26.17 | 27.91 | 57.96 |
| MotionCLIP [24] | B-actions | ✗ | - | 40.90 | 57.71 | - | - | - |
| TMR | B-actions | ✗ | 25.14 | 40.21 | 62.99 | 20.61 | 37.27 | 55.93 |
| TMR | B-text (raw) | ✗ | 25.36 | 37.93 | 54.14 | 20.88 | 34.03 | 47.95 |
| TMR | B-text (proc) | ✗ | 24.73 | 40.91 | 56.63 | 20.88 | 38.15 | 50.93 |
| TMR | H-text | ✗ | 22.44 | 27.10 | 53.73 | 16.23 | 23.66 | 44.67 |
| TMR | H-text | ✓w/o avg | 25.02 | 33.46 | 62.75 | 20.10 | 29.59 | 55.11 |
| TMR | H-text | ✓ | **26.30** | **36.08** | **64.18** | **22.20** | **32.46** | **56.32** |

Table 3. **Motion-to-text retrieval for action recognition:** Best results on BABEL action recognition in the *zero-shot* setting (last 3 rows) are obtained when training on HumanML3D (H-text) with all the text augmentations. We also provide results with the fully-supervised setting using action labels (B-actions). Benchmarking TMR [16] on this task obtains comparable performance to the state of the art. Finally, we report the intermediate setting of using raw or processed (proc) BABEL textual labels (B-text), from which action labels are inferred.

the evaluation of BABEL motion retrieval (i.e., large fluctuation when repeating the same experiment), we provide average over three repeated runs with different random seeds and report the standard deviation. Given the high variance on BABEL, we refrain from making conclusions on this new benchmark, but find its action retrieval evaluation to be more stable (see Section 4.4).

**Combining datasets.** Jointly training on HumanML3D and KITML (H+K) outperforms training only with one or the other when testing on the small-vocabulary KITML dataset. This does not impact performance on the larger HumanML3D. Adding BABEL to training does not bring a consistent boost, and mainly helps the same-domain BABEL evaluation.

**Text augmentation.** Text augmentations bring an overall improvement, especially significant on HumanML3D (14.47 vs 11.68 R@1). On the other hand, the impact on BABEL is inconclusive due to large variance in the BABEL retrieval benchmark. As will be seen in Section 4.4, the BABEL action recognition benchmark highly benefits from text augmentations. For more details on text augmentation parameters, we refer to Section 4.5.

**HN-NCE.** When replacing the InfoNCE loss with HN-NCE [19], we observe the best performance for H+K joint training when tested on HumanML3D and KITML. The best results on BABEL are also with HN-NCE, but when training on H+K+B.

To the best of our knowledge, these results represent state-of-the-art performance, with 3% improvement on HumanML3D over TMR [16] (11.63 vs 14.89), and with 7% improvement on KITML (21.75 vs 29.39).

### 4.4. Zero-shot action recognition results

We study the ability of a model trained on text labels, here HumanML3D, to generalize to categorical action labels, when evaluating on BABEL action recognition through motion-to-text retrieval. Following the original work describing the dataset and the action recognition benchmark [18], we report Top-1 and Top-5 accuracy metrics (equivalent to R@1 and R@5), as well as Top-1 class-balanced version (Top-1 CB). Results are summarized in Table 3. In the first block, we list the previous works reporting on this benchmark [18, 24], using the BABEL action labels for training (B-actions). We first check that TMR reaches their performance on this fully-supervised setting. We then provide intermediate results by using BABEL motions, but their free-form textual labels, instead of the categorical action labels. Both 'raw' and 'proc' (processed) labels provided by this dataset match the performance of action labels (perhaps due to action labels being derived from those). In the last block, we report the zero-shot setting by training on HumanML3D texts. Here, we observe significant improvements via text augmentations (e.g., 22.44 vs 26.30). We also ablate our average embedding strategy described in Section 3 ($p_{gt} = 0.4$, $p_{par} = 0.3$, $p_{act} = 0.3$, $p_{avg} = 0$) and see its benefits (last two rows).

### 4.5. Text augmentation ablations

We first study the impact of the choice of probabilities used in our augmentation strategy, $p_{par}$, $p_{sum}$ and $p_{act}$. Next, we compare our text augmentation approach to the one of Action-GPT [11], the method we find to be most related to ours. We conduct these ablations by training on the combination of HumanML3D + KITML training, and by evaluating on HumanML3D.

**Augmentation probabilities.** Table 4 studies the impact of the probability used for picking the augmentation approach when sampling the text label, among which are picking the ground truth ($p_{gt}$), picking one paraphrase ($p_{par}$), picking the action-type label ($p_{act}$), and picking the average of a random

| $p_{gt}$ | $p_{par}$ | $p_{act}$ | $p_{avg}$ | HumanML3D | | |
|---|---|---|---|---|---|---|
| | | | | R@1 | R@3 | R@10 |
| 1.0 | ✗ | ✗ | ✗ | 11.36 | 21.15 | 40.24 |
| ✗ | 1.0 | ✗ | ✗ | 13.23 | 24.34 | 42.43 |
| .6 | .4 | ✗ | ✗ | 13.30 | 24.48 | 45.12 |
| .4 | .6 | ✗ | ✗ | 13.37 | 25.66 | 44.87 |
| ✗ | ✗ | ✗ | 1.0 | 12.39 | 22.42 | 41.79 |
| .6 | ✗ | ✗ | .4 | 13.39 | 24.68 | 43.80 |
| .4 | ✗ | ✗ | .6 | 13.75 | 25.00 | 45.00 |
| .4 | .4 | ✗ | .2 | 13.30 | 24.32 | 44.75 |
| .4 | .2 | ✗ | .4 | 13.66 | 24.29 | 45.69 |
| .4 | .2 | .2 | .2 | 13.62 | 25.11 | 45.94 |
| .4 | .2 | .1 | .3 | 14.67 | 24.27 | 44.34 |

Table 4. **Ablations for text augmentation probabilities:** We train on the combination of HumanML3D and KIT, and investigate the impact of augmentation probabilities on the HumanML3D evaluation. While the model is not sensitive to the choice of these values, setting any of the 4 label types to zero (✗) reduces performance. The last row corresponds to H + K with augmentations in Table 2, where the mean across 3 runs is reported as 14.47 R@1.

| Averaging | $p_{gt}$ | $p_{avg}$ | $k$ | HumanML3D | | |
|---|---|---|---|---|---|---|
| | | | | R@1 | R@3 | R@10 |
| Token | ✗ | 1 | 4 | 8.87 | 17.77 | 33.12 |
| | | | rand/30 | 11.70 | 21.10 | 39.69 |
| Token | .5 | .5 | 4 | 11.79 | 20.92 | 39.53 |
| | | | rand/30 | 11.75 | 22.22 | 42.91 |
| Sentence | ✗ | 1 | 4 | 10.97 | 19.37 | 36.20 |
| | | | rand/30 | 12.32 | 22.70 | 41.51 |
| Sentence | .5 | .5 | 4 | 12.36 | 21.72 | 39.83 |
| | | | rand/30 | 14.03 | 24.50 | 43.61 |

Table 5. **Comparison to token averaging as in Action-GPT [11]:** We systematically analyze the impact of averaging multiple paraphrases of the textual label. Action-GPT performs token averaging before passing through the text encoder using a fixed number of $k = 4$ paraphrases, and does not use the original ground truth (GT) label. In our setting, averaging the *sentence* embeddings after the text encoder, for a random subset of a larger set of 30 paraphrases, using both GT and this average, outperforms significantly over this baseline (green vs red rows).

subset of labels ($p_{avg}$). Rows 2-4 experiment only with the paraphrasing approach, rows 5-7 only with the averaging approach, and rows 8-9 studies combinations of both, without including the action-type labeling approach. Finally, last two rows report combinations of these 3 approaches. While the model does not seem sensitive to the choice of the probability values, its performance increases when using a combination of all the augmentation approaches. We also observe that giving more weight to the averaging protocol further boosts the performance.

**Comparison to Action-GPT.** We compare our text augmentation to an approach we implement similar to Action-GPT [11]. Although used with a different training dataset, BABEL, on a different task (text-to-motion synthesis), this is the method we find to be most related to ours. More specifically, we compare both our ways of leveraging the use of several paraphrases for one text. Results are summarized in Table 5.

There are three main differences between our approach and the augmentation employed by Action-GPT: (1) For each text, they systematically generate a fixed amount ($k = 4$) of paraphrases, while we sample several texts at random from a larger paraphrases pool, i.e., random from 30. (2) They only use the paraphrased versions, but not the original label, i.e., $p_{gt} = 0$. (3) They average the paraphrase *tokens* at the entrance of the text encoder, while we average the *sentence* embeddings obtained after passing them through the text encoder (see Figure 2). Table 5 ablates each of these combinations, contrasting the approach of [11] that corresponds to the first row, with that of ours (last row).

Averaging the sentence embeddings performs clearly better than averaging the token embeddings for every parameter combination. We also validate our random sampling strategy, showing both the benefits of also including the ground truth labels, as well as not fixing the number of paraphrases.

## 4.6. Qualitative analyses

In this section, we provide visual illustrations of results for both text-to-motion retrieval (Figure 3) and action recognition (Figure 4). We further analyze action recognition results, in particular investigating per-action performances with/without text augmentations (Figure 5) and the confusions between actions (Figure 6).

Figure 3 shows qualitative results for text-to-motion retrieval on the HumanML3D test set, using the model trained on HumanML3D + KITML. We display two text queries, and top-5 ranked motions for each of them both with and without text augmentations. We notice that our model allows the retrieved motions to capture more elements and details of the input text. For instance in the above example, while the baseline captures the rough information that the query text targets the legs, the model with text augmentation captures the more specific interaction between knee and elbow in 4 motions out of 5.

In Figure 4, we illustrate several examples for the action recognition results on BABEL-60. We notice that while the correct action class is not always at the top rank, it often appears within the top 5 retrieved action labels. We observe that all top retrieved predictions are often related to the ground-truth action (e.g., 'Place something' vs 'Interact with/use object').

Figures 5 and 6 provide further insights, inspecting the per-class performances. Specifically, Figure 5 plots the R@1 score for each action before and after the text augmentations

Text query: 🔍 *A person lifts each knee towards the opposite elbow*

**Trained with augmentations**

1 (0.91)
*a man lifts each knee to his elbow multiple times and then does a squat.*

2 (0.88)
*a person lifts each knee towards the opposite elbow*

3 (0.87)
*a person standing forward doing leg kicks*

4 (0.84)
*a person touches each elbow to the opposite knee then spreads his legs and starts to do squats*

5 (0.84)
*person brings right elbow to left knee, then left elbow to right knee, stands straight then bends at the knees a few times*

**Trained without augmentations**

1 (0.88)
*a person standing forward doing leg kicks*

2 (0.87)
*a person kicking left leg and then kicking right out in front of them*

3 (0.87)
*a person sways to swing their right foot followed by their left foot*

4 (0.87)
*a man kicks with his right leg and then kicks with his left leg*

5 (0.85)
*a person lifts and spins around their right leg then lifts and spins around their left*

Text query: 🔍 *The man puts the box down and runs*

**Trained with augmentations**

1 (0.83)
*the man puts the box down and runs*

2 (0.83)
*a person, while running quickly, bends down and picks something up.*

3 (0.81)
*person jogs, stops to bend over, then continues jogging.*

4 (0.81)
*person runs quickly straight forward*

5 (0.80)
*a man crouches down while quickly walking forward and then stands up straight.*

**Trained without augmentations**

1 (0.85)
*this person does a short sprint forward, holding their arms up to their chest level*

2 (0.84)
*the person goes for a short jog*

3 (0.81)
*a person who is running*

4 (0.80)
*a person runs to the right slightly*

5 (0.80)
*the person is running forward*

Figure 3. **Qualitative results on HumanML3D text-to-motion retrieval with and without augmentation:** In both examples, while none of the retrieved motions are extremely remote from the text description, the model trained with augmentation captures more of the requested details for most motions in the top 5 ranks. In the example above, the model captures the interaction between elbow and knee, while the baseline model only captures the implication of the legs. In the below example, the model retrieves both parts of the movement – putting the box down and running – while the baseline only retrieves the running portion.



Figure 4. **Qualitative results on BABEL action recognition:** We apply zero-shot action classification via motion-to-text retrieval by treating class labels as text. The model is trained on HumanML3D free-form textual labels, and tested on BABEL actions. On the right of each input motion example, we display the ground truth (GT) action, along with the top-5 retrieved actions and their motion-text similarity scores. We observe that the high similarities among the top retrieved actions are mainly due to ambiguities across categories, e.g., "Grasp object" motion retrieves action classes involving hand motions such as "Touch object" and "Hand movements".

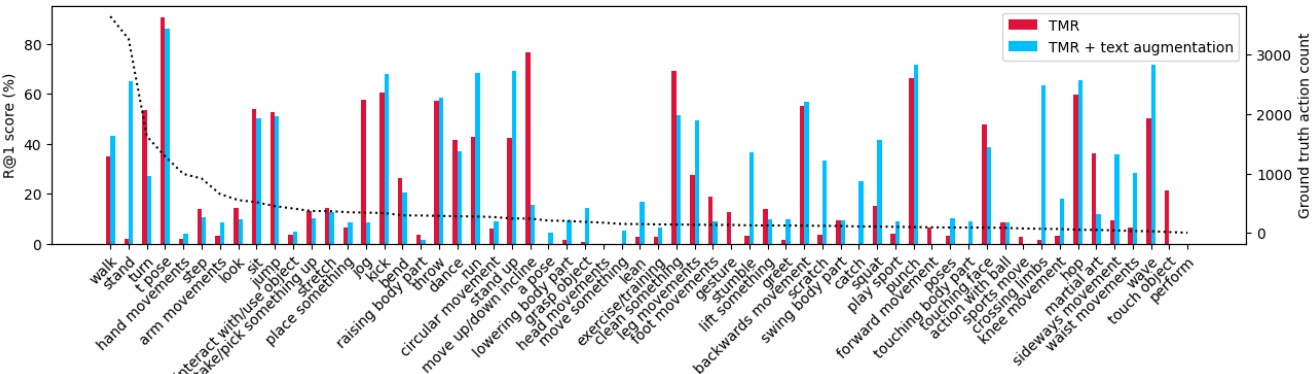

Figure 5. **Per-action performance improvement:** We plot the per-action R@1 scores for the 60 BABEL actions, comparing with/without the text augmentations. The dashed line represents the frequency of test labels for each class (y-axis on the right), showing the unbalanced nature of this benchmark.

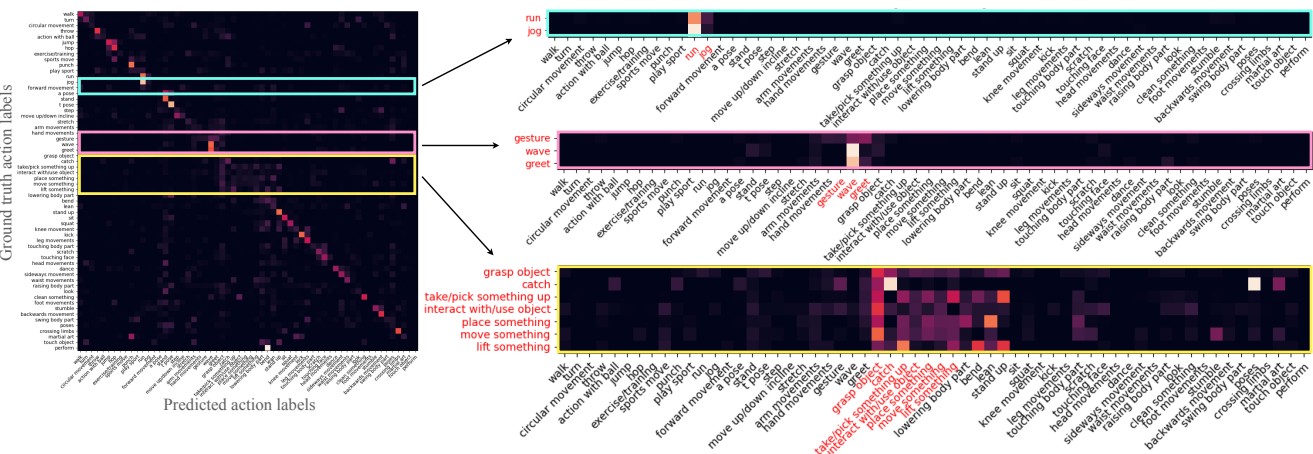

Figure 6. **Action classification confusion matrix portions:** We visualize several sources of classification mistakes, easily explained by the presence of ambiguous or related action labels. On the left, we display the full 60-categories of BABEL-60, and zoom into interesting regions on the right, highlighting the most confused actions in red. For example, the bottom row shows that hand-object interaction categories are confused frequently.

when training with the HumanML3D dataset. We observe that many more classes show a significant improvement than a loss of performance. For example, the rare classes in BABEL such as 'crossing limbs', 'wave', and 'knee movement' are substantially improved, as well as the frequent 'stand' category. Figure 6 further shows the most frequent confusion between categories, which demonstrates the finegrained nature of this benchmark. This allows to ponder the importance of some of the classification mistakes, by looking at the category an action is most confused with. As already outlined with Figure 4, some actions tend to be mostly mistaken for an action with similar meaning. For instance, the action 'jog', is mostly confused with 'run', which mitigates the fact that the performance of our model drops significantly on 'jog'. We also point in the confusion matrix a wide area corresponding to actions all related to hand-object interaction.

## 5. Conclusion and Limitations

We presented our work analyzing the generalization performance of text-motion retrieval models. Specifically, we perform cross-dataset experiments using standard benchmarks. Our results suggest that significant gains are observed when applying text augmentations to overcome the domain gap across datasets. Moreover, we benchmarked the popular TMR model on BABEL action recognition evaluation, and obtained promising zero-shot performance by only training on HumanML3D dataset. One potential limitation of our approach is the text augmentation which is not necessarily grounded in the motion. That is, the LLM can hallucinate details which are not visible in the motion. Future work can explore motion captioning as a way to incorporate grounded augmentations. Another avenue for future research is to expand this analysis to investigate the domain gap across motions, and not only across textual labels.

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
