# OpenReview forum: "A Cross-Dataset Study for Text-based 3D Human Motion Retrieval"
_thecvf.com/CVPR/2024/Workshop/HuMoGen — CVPR 2024 Workshop HuMoGen Submission_

### Official Review · Reviewer_8PHt · 2024-03-25
**accept**

**Rating:** 4
**Confidence:** 4

**Review:**

The authors point out the domain gap between the textual annotations of the three common text-motion datasets - BABEL, KIT, and HumanML3D. They suggest bridging the gap using textual augmentations with LLMs (namely, Llama-2 and GPT3.5). They use the augmented datasets to improve the motion retrieval task, retrain the recent TMR, and show improved results.

Pros
- Analyzing an important gap in the common text-motion datasets.
- Use it to improve motion retrieval and classification tasks.
- Comprehensive experiments - demonstrating the usefulness of the augmentation for single and multi-dataset training.

Cons
- The technical novelty is marginal.


Questions and suggestions for the next revision
- While you showed the domain gap in the textual annotations between the different datasets, you didn’t address the domain gap in the motion modality. Since both BABEL and HumanML are composed of subsets of AMASS and KIT is included in AMASS, there should be an overlap between the datasets. Measuring this overlap can be beneficial to understanding the usefulness of multi-dataset experiments.
- Please publish the textual augmentations for the datasets. That can be super useful for other text-to-motion tasks.

Overall, the paper highlights an important issue in text-motion research and shows it can be overcome with simple textual augmentations. This can be useful for the community, hence I would recommend accepting the paper.

---

### Official Review · Reviewer_88px · 2024-04-01
**This paper introduces a novel cross-dataset evaluation for the task of text-to-motion retrieval.**

**Rating:** 4
**Confidence:** 4

**Review:**

**Overview**

This work examines the generalization capability of a text-to-motion retrieval approach on three popular and publicly available 3D text-to-motion datasets. This work also attempts to unify these datasets by augmenting the input text, allowing their model to generalize beyond the original dataset they’re trained on. This study suggests that biases exist between the datasets. Furthermore, this work also demonstrates the zero-shot generalization capability of their proposed approach when tested on a dataset that is not used as a part of the training data.

**Strengths**

- The paper offers an extensive and interesting analysis of the popular 3D text-to-motion datasets which will be useful to the motion synthesis community.
- Once released, the cross-dataset data augmentation will be appreciated by the 3D human motion synthesis community, as it enables a larger training set for their model. This also opens a wider range of labeling options, enabling synthesis from either free-form text descriptions or shorter action-based labels across different datasets.
- It introduces a unified representation of the 3D motion sequences across different datasets.
- Their experimental analysis is quite extensive. The proposed modifications to the TMR text-to-motion retrieval model show positive improvements over the original design in general. These proposed components are also well ablated in their evaluation.

**Weaknesses**

- The proposed improvements over the TMR model appear minimal, as they mainly come from adjusting a contrastive loss function as well as the additional augmentation labels.
- The evaluation indicates that the proposed improvements for the text-to-motion retrieval task do not consistently guarantee better performance on datasets other than HumanML3D. However, this is also understandable due to the potentially high label-to-motion variance of certain datasets.
- The paper does not provide further discussion regarding dataset bias beyond what is presented in the introduction.

**Justification**

This paper presents a fresh perspective on how to analyze, examine, and dissect some of the overlooked aspects of the field of 3D human motion generation. In addition to being able to improve upon a state-of-the-art method of text-to-motion retrieval, it also offers a new upgrade on the popular 3D text-to-motion datasets, which will certainly be appreciated by the community.

---

### Meta-Review · Area_Chair_Q5sL · 2024-04-05

**Recommendation:** Accept

**Metareview:**

The paper received positive reviews. The authors appreciated the comprehensive evaluations presented in the draft and the interesting use of LLMs in the context of motion synthesis. The AC agrees with the reviewers and believes the paper would be highly relevant to the motion synthesis community. The authors are encouraged to release the data.

---

### Decision · Program_Chairs · 2024-04-06

**Decision:**

Accept

**Comment:**

The paper will be published as part of the official CVPR workshop proceedings upon submission of the camera-ready version.